# SAR ATR for Limited Training Data Using DS-AE Network

**DOI:** 10.3390/s21134538

**Published:** 2021-07-01

**Authors:** Ji-Hoon Park, Seung-Mo Seo, Ji-Hee Yoo

**Affiliations:** Agency for Defense Development, Daejeon 34186, Korea; seungmos@gmail.com (S.-M.S.); sich@add.re.kr (J.-H.Y.)

**Keywords:** synthetic aperture radar (SAR), automatic target recognition (ATR), deep learning, convolutional neural network (CNN), channel attention, double-squeeze-adaptive-excitation network, limited labeled data

## Abstract

Although automatic target recognition (ATR) with synthetic aperture radar (SAR) images has been one of the most important research topics, there is an inherent problem of performance degradation when the number of labeled SAR target images for training a classifier is limited. To address this problem, this article proposes a double squeeze-adaptive excitation (DS-AE) network where new channel attention modules are inserted into the convolutional neural network (CNN) with a modified ResNet18 architecture. Based on the squeeze-excitation (SE) network that employs a representative channel attention mechanism, the squeeze operation of the DS-AE network is carried out by additional fully connected layers to prevent drastic loss in the original channel information. Then, the subsequent excitation operation is performed by a new activation function, called the parametric sigmoid, to improve the adaptivity of selective emphasis of the useful channel information. Using the public SAR target dataset, the recognition rates from different network structures are compared by reducing the number of training images. The analysis results and performance comparison demonstrate that the DS-AE network showed much more improved SAR target recognition performances for small training datasets in relation to the CNN without channel attention modules and with the conventional SE channel attention modules.

## 1. Introduction

As the images from the synthetic aperture radar (SAR) can be acquired in all-weather and day-and-night conditions, they are widely used for surveillance and reconnaissance in the civil and military fields. As the numbers of operating SAR sensors and the images to be interpreted have increased, automatic target recognition using SAR images (SAR-ATR) has been one of the most important research topics in the last 30 years. According to [1], the recognition methods for SAR-ATR are classified into three taxa: feature-based taxon, model-based taxon, and semi model-based taxon. Among these, feature-based methods that include the template matching [2,3] and the pattern recognition [4,5,6,7,8,9] have brought more research attention in the SAR-ATR field. In recent years, deep learning-based methods such as the convolutional neural network (CNN) have rapidly replaced conventional pattern recognition methods as they dramatically improved the performances by automatically learning the discriminative features for SAR target recognition. In most relevant studies, excellent SAR target recognition performances were achieved by the combination of the CNN and pattern recognition [10,11,12,13], extraction of multi-view features [14,15,16], adoption of state-of-the-art deep learning network structures [17,18,19,20,21,22], fusion of feature maps [23], data augmentation [24,25], and transfer learning [26,27].

Unlike optical images, SAR target images formed by electromagnetic scattering phenomena are highly sensitive to changes in target poses or aspect angles. Therefore, in order to construct the deep learning networks for SAR-ATR with high performances, it is necessary to collect labeled target images with a variety of aspect angles and apply them to the network training. In practice, however, the SAR measurement process has a high cost and targets of interest may be located in inaccessible areas. This means that the quantity of the images would be limited to train the CNN and thus lead to performance degradation. Several studies have concentrated on the improvement of target recognition performance under this limited training data condition by approaching from a variety of perspectives. For example, in [28], Lin et al. proposed convolutional highway unit networks (CHU-net) that introduce adaptive gating algorithms to existing CNN structures. By designing an ensemble network using two CHU-nets, its inference resulted in a recognition rate near 95% using about 30% of training images. In [29], Zhang et al. applied the multi-view sequence of SAR target images to a bidirectional long short-term memory (Bi-LSTM), which is a kind of recurrent neural network (RNN). Based on multi-view image sequence data, comprehensive spatial features from Gabor filters with various directions and local binary pattern encodings were combined for application to the multi-aspect-aware Bi-LSTM. The presented framework demonstrated robustness to limited labeled data situations. In [30], Cho et al. developed the CNN with multiple feature aggregation based on the consideration of SAR image characteristics. Starting from the common feature map from a sequence of convolutional layers of CNN, it is separately operated with two different feature extraction paths. Then, the output feature maps with different scales are aggregated into a composite feature vector that is finally used for softmax classification. It showed a recognition rate of more than 90% in the case of 20% of training images. Similar to the approach in [30], Yu et al. presented the fusion of multi-scale feature maps derived from different layers of the CNN in [31]. By augmenting an input SAR image using multi-scale convolutional layers initialized by Gabor filters with different scales and directions, the resultant image cube is then passed through the CNN whose output feature maps from different network branches are combined before the classification process. It also achieved a recognition rate of more than 90% for 20% of training images. More recently, some studies have introduced new schemes for deep learning specialized to limited labeled data situations. In [32], Wang et al. proposed the few-shot learning for semi-supervised learning where support and query datasets of simulated SAR images help improve the recognition performance of unlabeled real SAR images. However, this study assumed that in spite of limited labeled data, other unlabeled data are relatively sufficient for guiding the decision boundary for classification. Thus, this is beyond the scope of this article where the quantity of target images for training is limited. In [33], Wang et al. proposed a few-shot learning scheme for SAR ATR based on a CNN for embedding feature projection, and an enhanced loss function for a hybrid method between inductive and transductive inferences. In [34], they proposed another approach for the few-shot learning scheme where a new network structure based on Bi-LSTM is applied to the feature map derived from the CNN. Apart from the few-shot learning, methods for supplementing SAR target images have also been proposed, recently aided by generative networks. In [35], Song et al. introduced zero-shot learning using the virtual SAR target images from a generative adversarial network (GAN). In [36], Cao et al. developed a label-directed GAN (LDGAN) based on the Wasserstein distance, and generated more realistic SAR target images to help improve the recognition performance in the situation of data shortage. Although they showed a promising potential in the case of extremely limited training images, the performance for real measured SAR images needs to be further improved. The aforementioned studies have dealt with the problem of limited training data in SAR-ATR from the viewpoint of network structures, multi-view image sequences, feature map fusion, and learning schemes, respectively. 

From the viewpoint of network structures, this article proposes a double squeeze-adaptive excitation (DS-AE) network to address the problem of limited training SAR target images. The DS-AE network is a kind of deep learning network consisting of a basic CNN such as a Residual Network (ResNet) [37] combined with the channel attention modules particularly designed in this article. According to the fundamental idea of the ‘squeeze-and-excitation (SE)’ network [38], which employs one of the most well-known channel attention methods, the channel attention mechanism can selectively emphasize the features useful for image classification and suppress the less useful ones included in the feature maps of the CNN. Due to this operation, the channel attention mechanism can have the potential to enhance the power of the CNN for representing discriminative features of SAR target images in spite of limited training data. However, because of two limitations of the SE network, the DS-AE network proposed in this article aims to further enhance the representation power of the SE network in the case of limited training images. First, in the squeeze stage of the original SE network, the channel information explicitly modeled by the global average pooling (GAP) operation suffers from drastic dimension reduction that may cause degradation of channel interaction, as noted in [39]. Thus, the DS-AE network has a double squeeze structure composed of one more fully connected layer (FC) for dimension reduction and one more FC for dimension recovery. This double squeeze operation acts as a ‘buffer’ that prevents a drastic change in the channel information and facilitates more channel interactions by augmented FC operations. Second, in the excitation stage of the SE network, the excitation vector is derived from the channel information activated by a fixed nonlinear function. Even though the tendency of using a fixed activation such as a rectified linear unit (ReLU) and sigmoid in the CNNs is highly common, introducing learnable parameters to the activation functions can bring a positive effect on improving the recognition performance, as noted in [40]. In this article, a new activation function referred to as parametric sigmoid (Psig) is devised to raise the adaptivity of the excitation vector, which is directly related to the discriminating capability of the feature map influenced by the channel attention mechanism. 

The main contribution of this article is twofold. First, to the best of the authors’ knowledge, this article attempts to address the problem of limited training images for SAR-ATR by the approach of channel attention in the CNN for the first time. Although a number of studies have presented applications of attention mechanisms to the problems of SAR target recognition [41,42], and SAR ship detection [43,44,45] in the meantime, the aforementioned problem has not been discussed. Second, the SE network is further improved from a network structural point of view by introducing the double squeeze operation and the adaptive activation function in the excitation process. Their advantages can be summarized as follows:Double squeeze: It has been reported that drastic dimension reduction of the channel feature vector can cause destruction of the direct correspondence between channels. By inserting additional FCs and activation functions, the double squeeze operation can prevent this drastic loss, thus leading to the enhancement of selective emphasis on the channel information.Adaptive excitation: Although existing channel attention methods use fixed activation functions, such as the sigmoid, this article newly introduces the parametric sigmoid whose translation and gradient are adaptively varied based on the dataset. By raising adaptivity in the excitation process, further performance improvement is expected for the limited training SAR images.Combination: The main factors of the DS-AE network, double squeeze and adaptive excitation, are complementary to each other. This effect was verified in the ablation study.

The remainder of this article is organized as follows. Section 2 explains ResNet18 modified for application to SAR-ATR as a basic CNN of the DS-AE network. In Section 3, a new channel attention module of the DS-AE network is described based on a brief review of the existing SE network. In Section 4, an ablation study using the moving and stationary target acquisition and recognition (MSTAR) dataset, where only 25% of training images are utilized for network learning, is performed with different network structures. Based on this study, analyses on how the channel attention of the proposed network improves the recognition performance are also provided. In Section 5, recognition performances are presented for different situations of limited SAR training images of the MSTAR dataset. Further comparison work with other previous studies is also briefly given in Section 5. Finally, Section 6 provides a conclusion of this article. 

## 2. Basic CNN of DS-AE Network

In this article, a Residual Network with 18 learnable layers (ResNet18) is employed as a basic CNN of the proposed DS-AR network. As described in [37], ResNet18 is characterized by skip connections that provide robustness to gradient vanishing of the deep CNNs. Here, this ResNet18 was modified for the application of SAR-ATR as follows:The input size of the network was changed to 128 × 128 × 1 (height × width × channel) considering the SAR target image as a kind of gray-scale image.The receptive field of the first convolutional layer with respect to the input image was changed from 7 × 7 to 3 × 3 considering the decreased size of the input image.No pooling operation such as the max-pooling or the average-pooling was included in this base CNN. Only convolutional operation with a stride of two was applied to size reduction of the feature maps.The number of nodes in the last FC was given as 10, which is the class number of the MSTAR dataset collected under standard operating conditions (SOCs).The pre-activation structure where the nonlinear activation (ReLU) is carried out before the convolutional operation was adopted to impose the attention mechanism onto the feature map from the residual connection, which is another path with convolutional units located parallel to the skip connection.

The whole architecture of the modified ResNet18 is shown in Figure 1. For better understanding of the network structure, the detailed layer composition of the input—first convolutional part—first half part of stage 1 is shown on the left side of Figure 1.

In Figure 1, there are 18 layers with learnable parameters and 8 skip connections that perform identity mapping by element-wise addition of the input feature map and the output of the stacked layers, namely the residual connection. For the input image with a size of 128 × 128 × 1, the first convolutional unit ‘conv 3 × 3 @64/s2p1+BR’ denotes that a two-dimensional convolutional layer with a kernel size of 3 × 3, a kernel number of 64, a stride of 2, and a zero-padding around the feature map (or input image) of 1 is followed by ‘BR’ that represents a sequence of a batch normalization layer and a ReLU nonlinear activation layer. By the zero-padding of the convolutional operation, the spatial size (height and width) of the output feature map becomes the original one divided by the stride number. The kernel number is directly related to the channel number of the output feature map. Thus, the output size of the first convolutional unit is 64 × 64 × 64, and this principle can be applied to other convolutional units. Subsequent convolutional units expressed by ‘BR+conv 3 × 3@C/s1p1’ have a different order of layer arrangement. As the ReLU activation is performed before the convolutional operation, it is referred to as the pre-activation. When investigating the overall network structure, there are four stages where the spatial sizes and the channel numbers of the feature maps decrease and increase, respectively. As shown in Figure 1, the first stage (stage 1) deals with the feature map with a size of 64 × 64 × 64 and the last stage (stage 4) processes that with a size of 8 × 8 × 512. That is, for the stage number of n, the spatial size and the channel number are $2^{7-n}$ and $2^{5+n}$, respectively. It is noted that the first convolutional layer in each stage has a stride of 2 that is related to spatial size reduction of the feature map. After passing through four stages, the input image is abstracted into a 512-element vector by the GAP operation, and then becomes a 10-element vector by the FC. This vector whose element number is the same as the number of SAR target classes of the MSTAR dataset is finally applied to the softmax nonlinear function. In the training step, the cross-entropy loss is minimized by the iterative comparison between the predicted vector from the softmax and the true label vector of the training SAR target image. In addition, L2-regularization is included to prevent the overfitting of network learning.

## 3. Channel Attention Module of DS-AE Network

### 3.1. Brief Review on SE Network

The SE network [38] is a network consisting of a basic CNN and SE channel attention modules with a representative channel attention mechanism. As any type of the basic CNN is available in the SE network, combining with ResNet18 is usually called SE-ResNet18. The SE channel attention module shown in Figure 2 is composed of two main steps. The first step is ‘squeeze’ where the input feature map with a size of *M* × *M* × *C* is encoded into a *C*-element vector through the GAP operation and two FCs with two nonlinear activations, ReLU and sigmoid. After computing the raw channel feature vector with a size of 1 × 1 × *C* via the GAP, its size is reduced by the first FC with *C*/*r* nodes. After activation with the ReLU, the channel feature vector is then recovered to its original size by the second FC with *C* nodes. Then, the sigmoid function acts as the final activation to the recovered channel feature vector to make an excitation vector with a size of 1 × 1 × *C*. For a reduction ratio of *r*, there is no fixed rule for its selection, which is dependent on applications. This article employs 16, as noted in the original paper of the SE network [38], where *r* of 16 was selected for balance between accuracy and complexity. Although *r* is less related to accuracy, its large value can relieve complexity by decreasing the node number of FCs in the SE channel attention module. The excitation vector resulting from the squeeze step can be regarded as numerically modeling the relative importance of individual channels by updating the parameters of two FCs. The second step is ‘excitation’ where the input feature map is multiplied by the result of the squeeze step (excitation vector) along the channel direction. Through this excitation step, the feature map channels with the discriminative information useful for SAR target recognition can be emphasized and those with relatively unimportant information are suppressed. In other words, ‘attention’ is performed by concentrating the attention of the network learning on the feature map channels with more important information. By this process, it is assumed that channel attention in the CNN is expected to improve the SAR target recognition performance under the limited training data as follows: 

From the explanation of the SE network, the channel attention mechanism can enhance the efficiency of the network learning by explicitly modeling the channel importance. This can lead the network to search for more discriminative features helpful for SAR target recognition.In the situation of limited training data, it is readily predicted that the performance of the network for target recognition becomes degraded especially for the SAR target images, which is highly sensitive to target poses and aspect angles. Nevertheless, the channel attention mechanism is expected to better capture the common features of target images belonging to the same category by the iterative learning of channel importance modeling. This assumption served as a major research motivation of this article.

### 3.2. Proposed Channel Attention Module of DS-AE Network

Although the SE network showed the enhanced representation power of the CNN in a number of relevant studies, each step of the channel attention process in the SE network has its own drawback to be improved. Thus, as described below, this article proposes a new channel attention module aiming to overcome the limitations of the existing SE network. Figure 3 shows the proposed channel attention module with the structures of double squeeze (DS) and adaptive excitation (AE). The DS is performed by the first eight layers (from the global average pooling layer to the fourth FC) and then derives a channel feature vector with a size of 1 × 1 × *C*. The AE starts from the application of the channel vector to the parametric sigmoid. Then, the activated channel vector, namely the channel excitation vector, is used for excitation of the input feature map by multiplying it along the channel direction.

In the squeeze step of the SE channel attention module, the channel feature vector derived from the GAP operation to the input feature map is compressed into a vector reduced by 1/*r* times in the number of elements. In the feature map, the interaction between its adjacent channels plays a role of providing discriminative information on image classification. However, as discussed in [39], it has been reported that this drastic dimension reduction of the channel feature vector may cause destruction of the direct correspondence between channels and lead to a decrease in the channel attention effect. To cope with this problem, the method without any dimensionality reduction based on the one-dimensional convolution was presented in [39]. Another method that circumvents the dimensionality reduction problem via the all-convolutional channel attention module was proposed in [42]. This article selects a different approach to gradually reducing and recovering the dimension of the channel feature vector by inserting two more FCs. As shown in Figure 3, the raw channel feature vector (1 × 1 × *C*) from the GAP is reduced into a *C*/*r*-element vector through the double FCs. Among these double FCs, the first one has 2*C*/*r* nodes to mitigate drastic loss in the channel information and the second one has the same number of *C*/*r* nodes as the node number of the single FC in the SE network. The gradually compressed vector with a size of 1 × 1 × *C*/*r* is then recovered to the original size in the reverse order of the previous reduction process. For activation of these vectors, the ReLU layer is located between each FC similar to the original SE network. In summary, the structure of ‘double squeeze’ is employed in the channel attention module of the proposed network to prevent drastic dimension reduction of the channel vector and to facilitate more interactions between channels by the augmented FCs.

Just before the excitation step of the SE channel attention module, the excitation vector directly applied to the input feature map is calculated from a fixed nonlinear activation function, which is the sigmoid, as presented in Figure 2. In most of the studies dealing with CNNs for image classification, there is a general tendency to use activation functions with fixed parameters. However, the introduction of additional learnable parameters to the activation functions can bring a positive effect on improving the recognition performance by increased adaptivity. In [40], thanks to a new variant of ReLU referred to as parametric ReLU (PReLU) whose learnable parameters control the gradient in the negative input domain, the ‘PReLU-Net’ became the first one to surpass the human-level image classification performance for the ImageNet dataset [46]. Motivated by this result, this article devises another new activation function, ‘parametric sigmoid (Psig),’ to raise the adaptivity of the excitation vector. The Psig adds two more learnable parameters *a* and *b* that have influences on the gradient and the translation of the original sigmoid (Figure 4) as follows:(1)$$\frac{e^{ax}}{b+e^{ax}}\;$$

In this article, the channel attention modules of the SE network and DS-AE network were inserted right before stage 1 and into element-wise addition layers of the basic CNN (modified ResNet18 in Section 2), as marked with the red dotted lines in Figure 1. It is noted that the channel attention modules were not inserted in the last stage of the basic CNN, as little performance improvement was expected for a feature map with the relatively large number of channels, as compared with the increase in the number of network parameters [38]. This is because, in the last stage, the excitation vectors from the channel attention module show weak or saturated responses that have less selective emphasis on channel information. By contrast, the last stage has a relatively large number of nodes, leading to a significant increase in the network complexity.

## 4. Ablation Study

In this section, the ablation study was carried out by applying the MSTAR dataset of SAR target images to five different types of networks listed below: Basic CNN without any channel attention modules;Basic CNN with SE channel attention modules (SE network);Basic CNN with SE channel attention modules whose squeeze operations are replaced by the double squeeze operations (DS-E network);Basic CNN with SE channel attention modules whose activation functions for computing the excitation vectors are replaced by the parametric sigmoid (S-AE network);Basic CNN with channel attention modules combined with 3 and 4 (DS-AE network).

### 4.1. MSTAR Dataset

The MSTAR public dataset acquired from the MSTAR project led by U.S. Air Force Research Laboratory (AFRL) is a well-known benchmark dataset for SAR-ATR of ground military targets. With the MSTAR dataset, most related studies developed their ATR algorithms. This article also used this dataset with 10 targets collected under the standard operating conditions, as summarized in Table 1. The targets were classified as several categories as follows: the tracked armored personnel carrier (BMP2), the wheeled armored personnel carrier (BTR70, BRDM2, and BTR60), the main battle tank (T72 and T62), the self-propelled howitzer (2S1), the anti-aircraft gun (ZSU234), the truck (ZIL131), and the bulldozer (D7). The SAR target images measured from a 17° depression angle were used to train the networks, and those from a 15° depression angle served as the test images applied to the evaluation of the trained networks. The optical images of these targets and corresponding SAR target images are depicted in Figure 5. 

As in the previous studies, only a fraction of the training images were used to simulate the situations of the limited training data. Here, ‘training ratio’ is defined as the ratio of the used training SAR target images to the whole training set. For example, when the training ratio is given as 50%, only half of the training images of each target class are randomly chosen and utilized for network learning. Note that all the test images were used for evaluation of the recognition performances.

### 4.2. Ablation Study with 25% of MSTAR Training Images

In the ablation study, five different types of network structures were evaluated with the MSTAR dataset described earlier. Each network structure (basic CNN, SE network, DS-E network, S-AE network, and DS-AE network) was trained with 25% of the training images of the MSTAR dataset. The recognition rate defined as the percentage of the test image correctly classified in the entire test dataset was employed as the evaluation metric. The networks were implemented using the Deep Learning Toolbox of MATLAB 2019b software, and they were trained from scratch based on the hardware with an Intel Xeon E5-2660 CPU and NVIDIA Geforce GTX 1080 GPU. The number of training images in a mini-batch is 32, and with this mini-batch scheme, stochastic gradient descent with momentum (SGDM) is adopted to minimize the cross-entropy loss of supervised learning. The image in a mini-batch are shuffled every epoch, which is the number of times the whole batch of the training images has been used during training. For a maximum epoch number of 200, the initial learning rate that determines the step size of the loss minimization is provided as 0.001, dropping by 50% at a period of 50 epochs based on the heuristic learning schedule. For data augmentation, the training images initially reduced to 25% are augmented by simple random pixel translations along the height and the width directions. It is noted that more sophisticated data augmentation methods such as noise addition and attributed scattering centers manipulation can be used for robust network training [24,47]. Considering that the SAR target images can be affected by uncertainty, several fuzzy preprocessing techniques used in other graphics fields may be required [48,49,50], although it is beyond the scope of the article focusing on the network structure.

Table 2 shows the result of the ablation study in terms of the recognition rates and the numbers of learnable parameters for five network structures. The SE network showed a recognition rate 8.6% larger than that of the basic CNN (modified Resnet18 with no channel attention). The DS-E network and the S-AE network exhibited slightly increased recognition rates. However, their combination, namely the DS-AE network, showed the much more improved recognition rate of 95.4%. It has been shown that two structural changes of double squeeze and adaptive excitation are complementary to each other and thus show a synergistic effect that is reviewed in the next subsection. In terms of the network size and efficiency, the parameter number of the DS-AE network increased by just 0.45% compared to that of the basic CNN and 0.27% compared to that of the SE network, respectively. For the training time, the networks with channel attention modules were inevitably time-consuming owing to the back-propagation training process through the increased layers that need high memory usage. However, there was little difference in inference time per one test image. Table 3, Table 4 and Table 5 show the confusion matrices with the detailed recognition results from the basic CNN, the SE network, and the DS-AE network. In the confusion matrix, each row denotes the actual target class, and each column represents the target class predicted by each network structure. ‘*P_cc_*’ denotes the recognition rate of correct classification with respect to each target class and the whole classes. Several observations from these confusion matrices are listed as follows: When the training ratio was given as 25%, numerous BMP2 target images were misclassified as the T72 target in the case of the basic CNN. This kind of phenomenon appears to originate from the resemblance between target configurations of BMP2 and T72, as discussed in [22]. When the channel attention modules were applied to the basic CNN, this problem was mitigated and was further improved in the DS-AE network.In the results from the basic CNN and the SE network, a number of BTR60 target images were confused by BTR70, which was included in the same category of the wheeled armored personnel carrier. However, by the enhanced representation power of the proposed DS-AE network, the number of misclassifications of BTR60 target images was substantially decreased.Although ZIL131 was the only target included in the category of trucks, a number of its target images were misclassified independently of categories, such as 2S1 (self-propelled howitzer), T62 (tank), and ZSU234 (anti-aircraft gun). The SE network was still confused in classifying between ZIL131 and T62. However, the DS-SE network showed more improved results in the classification of this target. In addition, for the D7 target, which was the only one included in the category of bulldozers, the DS-SE network had more discrimination capability in relation to the other two results in Table 3 and Table 4.Directly compared to the SE network and the DS-AE network, the latter showed higher or equal performances than the former for eight targets except for two targets, 2S1 and BRDM2. In the authors’ opinion, this misclassification of 2S1 may be caused by the fact that the appearance of 2S1 approximately resembles those of tanks, and its main body is built based on the armored personnel carriers. Thus, the DS-AE network is somewhat vulnerable to distinguishing 2S1 target images.

The most remarkable points of this ablation study can be summarized as follows:The approach of the channel attention mechanism in the CNN has high potential in solving the problem of performance degradation in the case of limited training data.The DS-AE network shows further improved recognition performance in relation to the existing SE network by its squeeze structure and adaptive excitation.

### 4.3. Further Analyses of Ablation Study

In this subsection, further works are presented for more in-depth analyses of why the proposed network improves the recognition performance using the class activation maps [51], and channel-wise activation maps [22,41]. The learned parameters of Psig controlling the original sigmoid shape and the effect of double squeeze and adaptive excitation on the excitation vectors are also discussed.

#### 4.3.1. Analysis with Class Activation Maps

Figure 6 shows the class activation maps of two example test images to give more insight into the classification performed by the networks. The class activation map is a useful tool for explainable artificial intelligence and helps identify which part of the SAR target image is responsible for the classification result. The first column describes the original SAR target images of T72 and BMP2 targets. From the second to the fourth columns in Figure 6, the class activation maps from three network structures trained with 25% of training images are overlapped with the SAR target images. In the first row exhibiting the test SAR target image of T72, the strong part of the class activation map from the basic CNN is deviated from the central target region, and thus the shadow region is mainly attributed to the classification result. The maps from the networks with the channel attention mechanism show a higher concentration on the target part. In the case of the SE network, however, the map is widespread throughout the image including the background region. This means that not only the target region but also the background contributes to the classification result. In several studies [17,52], the recognition performances were evaluated using the images with only target regions segmented to exclude the effect of the background region. On the contrary, the DS-AE network yields a class activation map more focused on the target region. For the second row of the BMP2 target, the map from the DS-AE network shows a high concentration on the target region and the shadow region that are both used for classification. In contrast, the other two networks yield maps deviated from the target region or widespread throughout the image, including the part unrelated to the target signature. This analysis using the class activation maps indirectly demonstrate that the DS-AE network has the advantage of SAR target recognition for limited training data.

#### 4.3.2. Analysis with Channel-Wise Activation Maps

For further analysis of the channel attention effect on performance improvement, channel-wise activation maps, the feature maps sliced along the channel direction, were compared for three different networks, as shown in Figure 7 and Figure 8. Note that due to space limitations, only the 16 strongest maps out of 64 channels at the end of the stage 1 (basic CNN in Figure 1) are provided for the target images used in the previous subsection. It is also noted that a more intuitive observation is possible in the shallow stage with a low degree of feature abstraction.

For all the networks trained with 25% of SAR training images, the channel-wise activation maps from the basic CNN (Figure 7a and Figure 8a) show relatively weak responses from the target part of the image, while the other maps affected by channel attention exhibit more apparent activation appearances. From the maps in Figure 7b and Figure 8b, the channel attention modules in the SE network enhance the responses from the target regions and those from the shadow regions, particularly in Figure 8b. This emphasis on the shadow region can be related to the distribution of the class activation map observed in Figure 6. Although the shadow responses are suppressed compared to the maps from the SE network, those from the DS-AE network show much stronger target responses related to further improvement of the recognition performance. This characteristic will be beneficial to other datasets of SAR target images where accurate identification of the shadow regions is not available. One more peculiar observation from the above channel-wise activation maps is that the weakest one of the 16 maps from the DS-AE network appears to be nearly blank. As discussed in [41], this indicates that the channel attention modules in the DS-AE network emphasize the important features and suppress the less useful ones more selectively than the original SE channel attention modules. The comparison of channel-wise activation maps provides additional evidences to support that channel attention is effective for improving the SAR-ATR performance under the condition of training data restricted to 25%.

#### 4.3.3. Discussion on the Learned Parameters of Psig

The parametric sigmoid (Psig) with two learnable parameters is defined for each element of the channel excitation vector. For example, at stage 1 of the network where a feature map has 64 channels, 64 Psigs have their own learned parameters *a* and *b* related to the gradient and the translation of the original sigmoid. In this subsection, the additional parameters of Psigs are checked after network learning of the DS-AE network. For the shape of the original sigmoid shown in Figure 9a, the shapes of the Psigs corresponding to the channels of the eight strongest responses (red lines) and eight weakest responses (blue lines) for the input test image of T72 are shown in Figure 9b–d. Figure 9b–d present the Psigs at the end of stage 1, stage 2, and stage 3 of the DS-AE network, respectively. As mentioned before, there is no Psig in stage 4. From Figure 9, all the Psig have different shapes from that of the original sigmoid by additional parameters to be learned. The most remarkable observation is that the ‘red’ Psigs shows more different aspects of shapes in relation to the ‘blue’ Psigs. Thus, it is assumed that the channels with the strong target responses have Psigs whose parameters change the original sigmoid to a greater extent. Figure 10 for the input test image of BMP2 presents a similar trend between the Psigs of strong and weak channel responses. The discussion of the parameters of Psig illustrates the adaptivity of the excitation process carried out in the DS-AE network. 

#### 4.3.4. Investigation into Excitation Vectors from Double Squeeze and Adaptive Excitation

In this last subsection, the channel excitation vectors from the output of the respective stages of the DS-AE network are investigated to analyze the effect of double squeeze and adaptive excitation on channel attention. Figure 11 shows the excitation vectors with the double squeeze operation only and with both double squeeze and excitation when the input test image of BMP2 target is applied to the DS-AE network. For the output from stage 1 shown in Figure 11a, the vector only with double squeeze (blue line) has values generally less than those of the vector with the single squeeze in the SE network (red line). In Figure 11b, depicting after adaptive excitation by the Psig, the difference between the emphasized and suppressed values becomes more discernible in the excitation vector from the DS-AE network (red line) than that from the original sigmoid of the SE network (blue line). The excitation vector from the SE network is just shifted to the response range between 0 and 1 by the original sigmoid. This observation result is closely related to the results of channel-wise activation maps. For the output from stage 2 shown in Figure 11c,d, the vectors from the DS-AE network (blue lines) present more discrimination capability between the channels in the case both before and after adaptive excitation. It is assumed that the double squeeze operation of this stage is relatively important for improving the SAR target recognition performance. For the output from stage 3 shown in Figure 11e,f, however, it is difficult to find the remarkable difference between the excitation vectors from the DS-AE network and the SE network. As discussed in [38], this observation is in accordance with the fact that small performance improvement is expected by channel attention for the feature map with a number of channels. In summary, the investigation into the excitation vectors can support the complementary and synergic relationship between double squeeze and adaptive excitation. When the double squeeze operation derives the vector with good discrimination between important and unimportant channel features, the resultant excitation vector from the Psig can have a more improved selectivity of channels. In contrast, if the double squeeze operation yields small responses, the adaptive excitation by the Psig can supplement it, as shown in Figure 11a,b.

## 5. Experimental Results for Various Conditions of Limited Training Data

Based on the ablation study, the DS-AE network was evaluated for various conditions of limited training data where the training ratios are given as 10%, 20%, 25%, 30%, 50%, and 100%, respectively. The hyper-parameters and other settings for network training are the same as those in the previous section. Table 6 presents the recognition rates of the network structures used in the ablation study, the basic CNN, the SE network, and the DS-SE network. When all the training SAR target images are used (100% of training ratio), not only the networks with channel attention modules but also the basic CNN can achieve a recognition rate more than 99% for the MSTAR dataset. Even if the DS-AE network shows the best performance of 99.30%, there is little difference between the networks with and without channel attention modules. This is because only a few learnable parameters are added to the basic CNN, which has many more parameters in the situation of sufficient training data. However, the fewer images used for network training, the more differences in target recognition rates can be observed between the CNN with and without the channel attention modules. In particular, the DS-AE network shows a recognition rate of more than 98% for the training ratio of 50% and 95.75% for the training ratio of 30%. When the training ratio decreases to nearly 20%, the performance of the DS-AE network remains at around 95% of the recognition rate contrary to that of the SE network. For the training ratio of 10% where the aspect angular interval between adjacent images is 13° on average, more than 80% of the test SAR target images are correctly classified, which is 14.3% and 6.7% higher than the basic CNN and the SE network, respectively. 

In Table 6, the recognition rates in previous studies are also presented for comparison. It is noted that these results were not reproduced here but cited as published in the references [28,29,30,31]. As all of these studies randomly selected a portion of training images to simulate the situation of limited data, the detailed composition of the training images in this article can be different from those of previous studies. Although reproduction of their results using the same training data may be desirable for more accurate performance comparison, their detailed implementation procedures are not available, and implementing them only with descriptions in those references may lead to distorted results. Therefore, Table 6 provides these recognition rates to indirectly show the feasibility and validity of the proposed DS-AE network and not to clarify its superiority to other methods. Here, there are three notations of the recognition rates:The notation ‘00.00%’ means the rate explicitly described in the references.The notation with the numerical range ‘~’ means the estimated value from the graph presented in the references where its numerical value could not be found.The notation ‘-’ means the rate not presented in the references.

When the training ratio is given as more than 50%, all the results from the DS-AE network and other studies present high recognition rates. Several previous results remain near 95% for the training ratios of 25% and 30%. However, for the training ratio of less than 20%, the performances are shown to be degraded by the limited training data. The experimental results in Table 6 demonstrate that the DS-AE network can achieve competitive performances compared to other studies, and can be an effective alternative for enhancing the performance for the case of limited training data.

## 6. Conclusions

In order to address the problem of SAR-ATR performance degradation for the limited training data, this article proposed a new network structure, a DS-AE network with a modified ResNet18 as the basic CNN and channel attention modules. The double squeeze structure in the proposed modules prevents drastic dimension reduction of the channel vector and then leads to more interactions between channels by augmented FCs. The adaptive excitation of the DS-AE network is performed by the parametric sigmoid that has more learnable parameters controlling the sigmoid function. From the ablation study, the analysis results showed that in the case of limited training data, the DS-AE network improved the SAR target recognition performance in relation to the CNN without the channel attention mechanism and the SE network. Further experiments with various conditions of limited training data and comparisons with other previous studies also demonstrated that the approach proposed in this article can have competitive performance. In conclusion, this article has shown two main contributions: by validating the effectiveness of channel attention to cope with limited training data and by presenting new network structures that enhance the existing SE network.

## Figures and Tables

**Figure 1 sensors-21-04538-f001:**
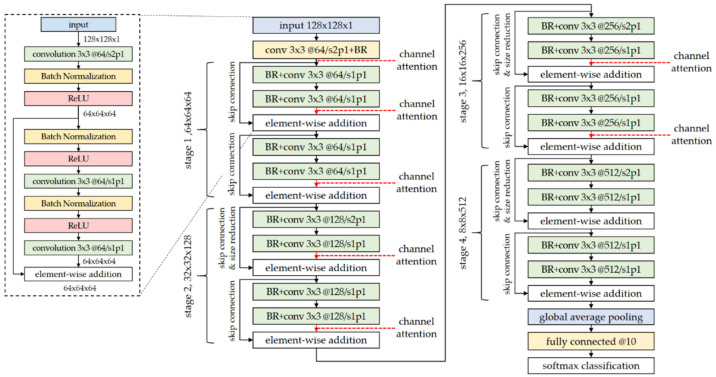
Structure of the ResNet18 modified for SAR-ATR application. The above ResNet18 has 18 layers with learnable parameters and 8 skip connections, identical to the original ResNet18. The main modifications are: the input size, the receptive field size of the first convolutional layer, the number of nodes of the last fully connected layer, and the adoption of the pre-activation structure.

**Figure 2 sensors-21-04538-f002:**
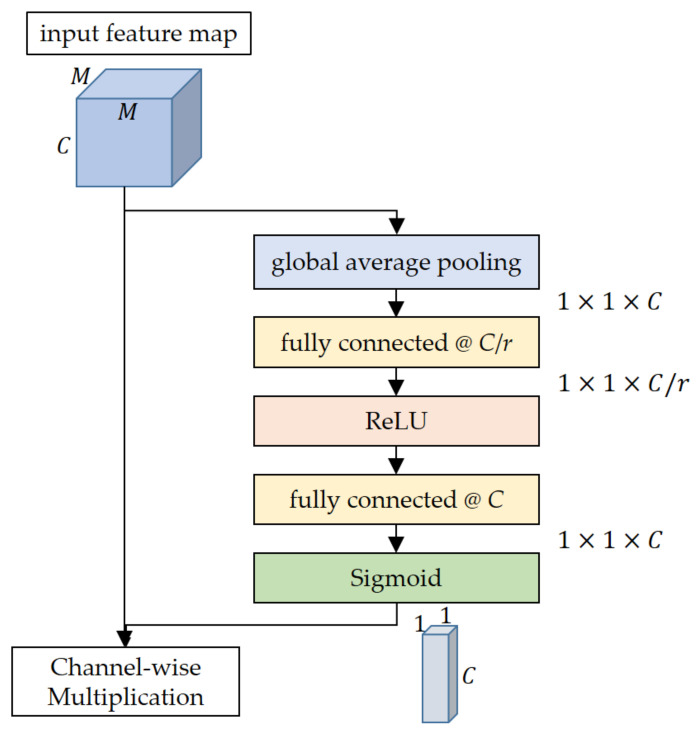
Structure of the SE channel attention module. The input feature map is squeezed and recalibrated via the global average pooling and two fully connected layers. Each fully connected layer is followed by the activation functions, the ReLU and the sigmoid.

**Figure 3 sensors-21-04538-f003:**
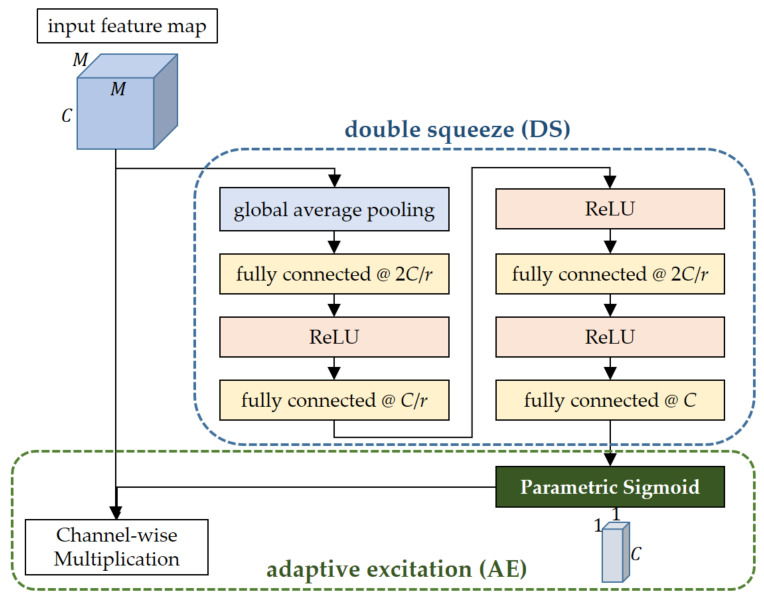
Structure of the channel attention module of the DS-AE network. It has the double squeeze structure implemented by four sequential fully connected layers where each one is activated by the ReLU except for the last one followed by the parametric sigmoid with more adaptivity compared to the original sigmoid.

**Figure 4 sensors-21-04538-f004:**
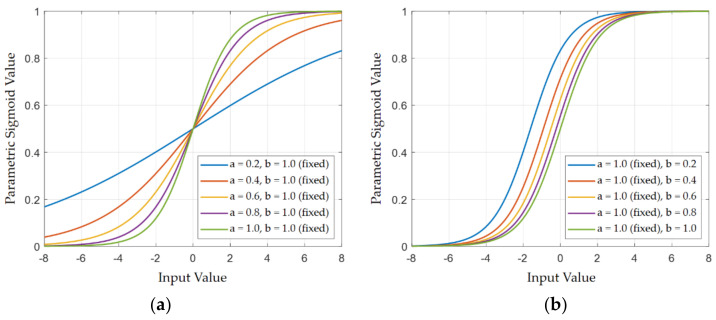
Shape of the parametric sigmoid (**a**) for different values of *a* and fixed *b* as 1, and (**b**) for different values of *b* and fixed *a* as 1. Note that both *a* and *b* are adaptively determined during the network learning process. When both *a* and *b* are 1, the parametric sigmoid is degenerated into the original sigmoid.

**Figure 5 sensors-21-04538-f005:**
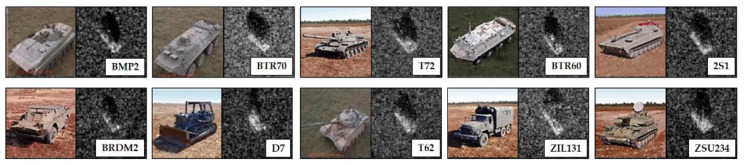
Optical and SAR images of 10 targets in the MSTAR dataset.

**Figure 6 sensors-21-04538-f006:**
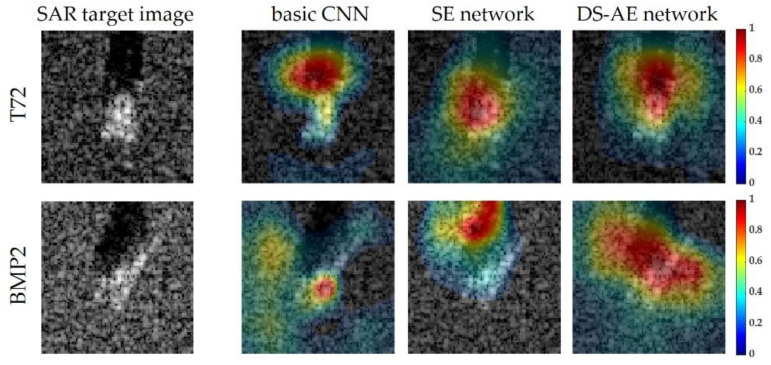
SAR target images and class activation maps of T72 (1st row) and BMP2 (2nd row).

**Figure 7 sensors-21-04538-f007:**
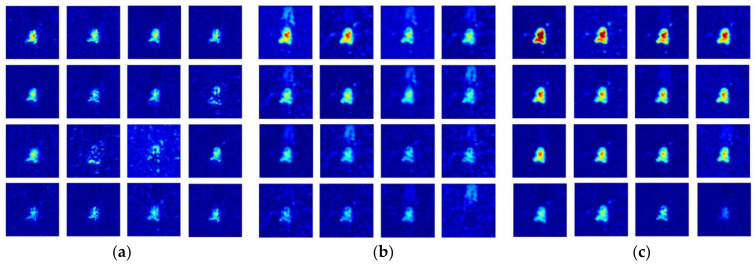
16 strongest channel-wise activation maps of T72 SAR target image. (**a**) Activation maps from the basic CNN; (**b**) activation maps from the SE network; (**c**) activation maps from the DS-AE network.

**Figure 8 sensors-21-04538-f008:**
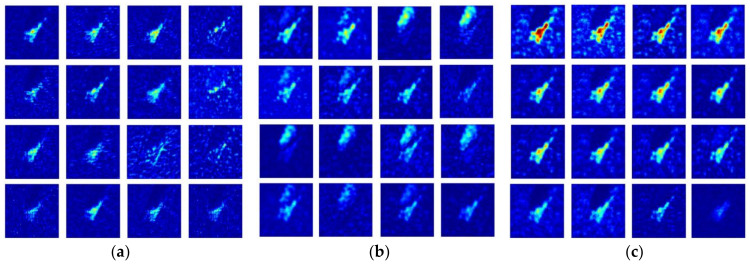
16 strongest channel-wise activation maps of BMP2 SAR target image. (**a**) Activation maps from the basic CNN; (**b**) activation maps from the SE network; (**c**) activation maps from the DS-AE network.

**Figure 9 sensors-21-04538-f009:**
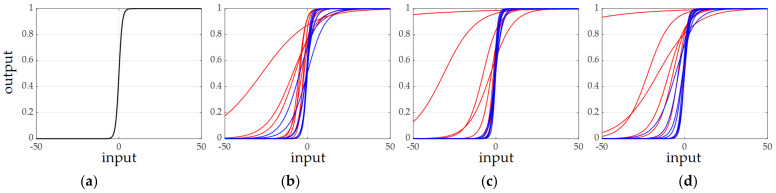
Shapes of the original sigmoid and the parametric sigmoids for the test image of T72 input to the DS-AE network learned by 25% of MSTAR training images. The red and blue correspond to the parametric sigmoids of channels with strong and weak responses, respectively. (**a**) Original sigmoid; (**b**) parametric sigmoids of stage 1 in the basic CNN; (**c**) parametric sigmoids of stage 2 in the basic CNN; (**d**) parametric sigmoids of stage 3 in the basic CNN.

**Figure 10 sensors-21-04538-f010:**
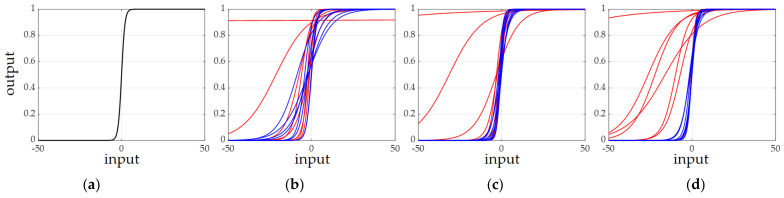
Shapes of the original sigmoid and the parametric sigmoids for the test image of BMP2 input to the DS-AE network learned by 25% of MSTAR training images. The red and blue correspond to the parametric sigmoids of channels with strong and weak responses, respectively. (**a**) Original sigmoid; (**b**) parametric sigmoids of stage 1 in the basic CNN; (**c**) parametric sigmoids of stage 2 in the basic CNN; (**d**) parametric sigmoids of stage 3 in the basic CNN.

**Figure 11 sensors-21-04538-f011:**
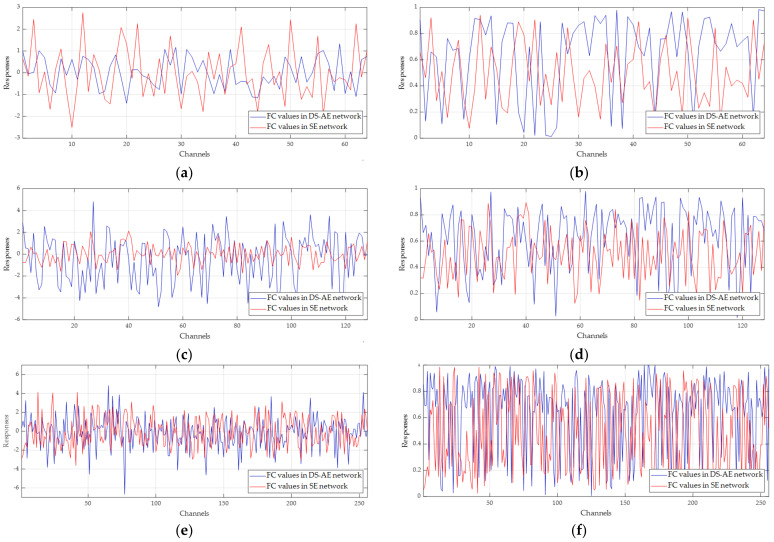
Channel excitation vectors from the DS-AE network (blue line) and the SE network (red line) for stages 1, 2, and 3. (**a**) Channel excitation vectors at the output of stage 1 before excitation; (**b**) channel excitation vectors at the output of stage 1 after excitation; (**c**) channel excitation vectors at the output of stage 2 before excitation; (**d**) channel excitation vectors at the output of stage 2 after excitation; (**e**) channel excitation vectors at the output of stage 3 before excitation; (**f**) channel excitation vectors at the output of stage 3 after excitation.

**Table 1 sensors-21-04538-t001:** Information on the SAR target images of MSTAR dataset collected under SOC.

Target Type	Serial Number	Training Images	Test Images
BMP2	9563	233	196
BTR70	C71	233	196
T72	132	232	196
BTR60	k10yt7532	256	195
2S1	b01	299	274
BRDM2	E71	298	274
D7	92v13015	299	274
T62	A51	299	273
ZIL131	E12	299	274
ZSU234	D08	299	274

**Table 2 sensors-21-04538-t002:** Result of the ablation study with 25% of MSTAR training images.

Network Structure	Recognition Rate	Parameter Number	Training Time	Inference Time
Basic CNN	85.70%	11.18 M	35 min	1.81 ms
SE network	94.23%	11.20 M	146 min	2.08 ms
DS-E network	94.40%	11.22 M	171 min	2.16 ms
S-AE network	94.64%	11.20 M	160 min	2.12 ms
DS-AE network	**95.42%**	11.23 M	189 min	2.18 ms

**Table 3 sensors-21-04538-t003:** Confusion matrix of the recognition result from the basic CNN related to Table 2.

Class	BMP2	BTR70	T72	BTR60	2S1	BRDM2	D7	T62	ZIL131	ZSU234	*P_cc_* (%)
BMP2	145	6	24	8	8	2	0	1	1	1	73.80
BTR70	4	169	2	9	8	3	0	1	0	0	86.22
T72	6	1	178	0	2	0	0	6	1	2	90.82
BTR60	0	15	2	167	0	1	3	5	0	2	85.64
2S1	0	10	0	17	222	1	0	21	3	0	81.02
BRDM2	23	0	0	3	9	230	0	0	6	3	83.94
D7	0	0	0	0	0	0	263	1	4	6	95.99
T62	0	0	4	2	1	0	2	237	5	22	86.81
ZIL131	0	0	1	0	14	2	3	16	204	32	74.45
ZSU234	0	0	0	0	1	0	5	1	3	264	96.35
Total											**85.70**

**Table 4 sensors-21-04538-t004:** Confusion matrix of the recognition result from the SE network related to Table 2.

Class	BMP2	BTR70	T72	BTR60	2S1	BRDM2	D7	T62	ZIL131	ZSU234	*P_cc_* (%)
BMP2	176	4	7	3	5	0	0	1	0	0	89.80
BTR70	4	185	1	5	1	0	0	0	0	0	94.39
T72	3	1	186	0	3	0	0	1	0	2	94.90
BTR60	1	14	12	163	1	0	0	0	0	4	83.59
2S1	0	4	7	0	259	0	0	3	1	0	94.53
BRDM2	1	0	0	0	1	272	0	0	0	0	99.27
D7	1	0	0	0	0	0	261	0	1	11	95.26
T62	0	0	2	0	1	0	0	264	0	6	96.70
ZIL131	1	0	0	0	6	1	1	15	247	3	90.15
ZSU234	0	0	0	0	0	0	1	0	0	273	99.64
Total											**94.23**

**Table 5 sensors-21-04538-t005:** Confusion matrix of the recognition result from the DS-AE network related to Table 2.

Class	BMP2	BTR70	T72	BTR60	2S1	BRDM2	D7	T62	ZIL131	ZSU234	*P_cc_* (%)
BMP2	184	0	4	1	5	1	0	1	0	0	93.88
BTR70	4	185	2	4	0	1	0	0	0	0	94.39
T72	2	0	186	0	1	0	0	4	3	0	94.90
BTR60	2	7	4	175	1	0	0	0	0	6	89.74
2S1	4	4	1	1	248	2	0	7	7	0	90.51
BRDM2	2	0	0	0	1	270	1	0	0	0	98.54
D7	0	0	0	0	0	0	268	0	2	4	97.81
T62	0	0	2	0	1	0	0	267	1	2	97.80
ZIL131	0	0	0	0	3	0	2	9	258	2	94.16
ZSU234	0	0	0	0	0	0	0	0	0	0	100.00
Total											**95.42**

**Table 6 sensors-21-04538-t006:** Comparison of SAR target recognition rates for different training ratios.

Network Structure	Training Ratio					
	10%	20%	25%	30%	50%	100%
basic CNN	67.93%	83.14%	85.70%	89.65%	94.02%	99.01%
SE net	75.43%	92.87%	94.23%	94.85%	97.28%	99.13%
DS-AE net	**82.19%**	**94.85%**	**95.42%**	**95.75%**	**98.06%**	**99.30%**
Lin [28]	~35%	~75%	-	94.97%	97~98%	99.09%
Zhang [29]	-	-	95.79%	-	97.74%	99.90%
Cho [30]	69.81%	91.08%	-	92.95%	94.85%	95.52%
Yu [31]	-	92.12%	-	~95%	~97%	99.83%

## Data Availability

MSTAR dataset: The Air Force Moving and Stationary Target Recognition Database. https://www.sdms.afrl.af.mil/datasets/mstar/.

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
