# Peer review of "SAR ATR for Limited Training Data Using DS-AE Network"

_sensors, 2021, doi:10.3390/s21134538_

Round 1

Reviewer 1 Report

*) Figures 1, 2 and 3 are very important. So, I think they must have self-explanatory captions.

*) The most significant numerical results reported in the Tables should be highlighted in bold.

*) The images analyzed, of course, could be affected by uncertainty. Hence the need to use fuzzy preprocessing techniques would arise. Notwithstanding that such an activity goes beyond the already valuable work of the Authors, I recommend inserting a sentence in the text that highlights this possibility by putting the following works in the bibliography:

doi: 10.1007/s40815-020-01030-5

doi: 10.1016/j.knosys.2019.105279

doi: 10.1007/s11042-020-08636-9

*) Please explain in more detail the structure shown in Figure 1.

Reviewer 3 Report

This article proposes a double squeeze-adaptive excitation (DS-AE) network for small sample problem in SAR ATR. I have the following comments:

  1. It is said that this manuscript addresses the SAR ATR by the approach of channel attention in the CNN for the first time. But in “Ship Detection in Large-Scale SAR Images Via Spatial Shuffle-Group Enhance Attention,in IEEE TGRS, 2021” said that, the SENet has performed attention operations on channels, which means that channel attention is not a novel work. If just use it to solve the limited data problem, it cannot be seen as a novel contribution.
  2. For small sample problem, there are some other solving methods, like sample generation via GNN, like “LDGAN: A Synthetic Aperture Radar Image Generation Method for Automatic Target Recognition, 2020”. Maybe there should be some comparison.
  3. This article proposes a new channel attention module aiming to overcome the limitations of the existing SE network. From Fig.2 and Fig.3, it seems like the proposed new channel attention module only adds two fully-connected layers and one ReLu? Why this structure can solve the limited data problem?
  4. In section 4.2, 25% training samples, how to choose this 25% samples? Randomly or manual selection with equal azimuth interval? Also in table 6, how to choose the samples with ratios of 10%-50%?
  5. The function of DS is clear, but the function of AE seems no related description. In Fig.3 or some other places should add the AE description.

Round 2

Reviewer 3 Report

Can be accepted.